# The Autonomous Parvovirus Minute Virus of Mice Localizes to Cellular Sites of DNA Damage Using ATR Signaling

**DOI:** 10.3390/v15061243

**Published:** 2023-05-25

**Authors:** Clairine I. S. Larsen, Kinjal Majumder

**Affiliations:** 1Institute for Molecular Virology, University of Wisconsin-Madison, Madison, WI 53706, USA; cilarsen@wisc.edu; 2Cellular and Molecular Biology Graduate Program, University of Wisconsin-Madison, Madison, WI 53706, USA; 3McArdle Laboratory for Cancer Research, University of Wisconsin School of Medicine and Public Health, Madison, WI 53706, USA; 4University of Wisconsin Carbone Cancer Center, University of Wisconsin School of Medicine and Public Health, Madison, WI 53706, USA

**Keywords:** parvoviruses, DNA damage response, Minute Virus of Mice

## Abstract

Minute Virus of Mice (MVM) is an autonomous parvovirus of the *Parvoviridae* family that replicates in mouse cells and transformed human cells. MVM genomes localize to cellular sites of DNA damage with the help of their essential non-structural phosphoprotein NS1 to establish viral replication centers. MVM replication induces a cellular DNA damage response that is mediated by signaling through the ATM kinase pathway, while inhibiting induction of the ATR kinase signaling pathway. However, the cellular signals regulating virus localization to cellular DNA damage response sites has remained unknown. Using chemical inhibitors to DNA damage response proteins, we have discovered that NS1 localization to cellular DDR sites is independent of ATM or DNA-PK signaling but is dependent on ATR signaling. Pulsing cells with an ATR inhibitor after S-phase entry leads to attenuated MVM replication. These observations suggest that the initial localization of MVM to cellular DDR sites depends on ATR signaling before it is inactivated by vigorous virus replication.

## 1. Introduction

The autonomous parvovirus Minute Virus of Mice (MVM) is lytic in murine hosts and in transformed human cells [1]. The viral genome is single-stranded with inverted terminal repeats (ITRs) at either end that serve as packaging signals and as origins of replication [2]. MVM expresses two non-structural proteins, NS1 and NS2. While NS1 is essential for virus replication, NS2 is required only in murine hosts [3,4,5,6,7,8]. MVM depends on host cell cycle entry into S-phase to initiate viral replication, utilizing host DNA polymerase delta and alpha to amplify its genome [9,10]. Virus replication proceeds via a partial strand displacement process known as rolling hairpin replication [11]. MVM establishes viral replication centers known as Autonomous Parvovirus-Associated Replication (APAR) bodies in the nuclear environment that colocalize with cellular replication and repair proteins [12,13,14]. Vigorous MVM replication leads to a potent pre-mitotic cell cycle block at the G2/M border mediated by transcriptional silencing of host Cyclin B1 [15,16]. 

It has become increasingly clear that DNA viruses associate with host cell DNA damage response (DDR) proteins in the nuclear environment [17,18]. These DDR proteins colocalize with viral replication centers. In particular, DNA viruses like HPV and HBV associate with persistent cellular DDR sites, known as fragile sites [19,20,21,22]. Cellular fragile sites are induced by collisions between the host cell replication and transcription machinery and are stabilized by host topoisomerases and cohesin [23,24,25]. We have previously discovered that MVM genomes localize to cellular DDR sites in order to establish efficient infection [26]. In particular, a subset of cellular fragile sites induced early in S phase (referred to as Early Replicating Fragile sites, or ERFs) are preferred regions for MVM to establish replication centers with the help of NS1 [27]. We have previously discovered that NS1 transports the viral genome to cellular DDR sites by binding to the viral genome and to heterologous DNA molecules containing NS1 binding elements [27]. However, the cellular signaling pathways that drive MVM localization (either NS1 or the viral genome) to cellular DDR sites remain unknown. 

DNA viruses modulate the cellular DDR pathways using distinct strategies for their benefit. The related parvovirus Adeno-Associated Virus 2 (AAV2) induces a cellular DDR that is driven by the PI3-kinase-like-kinase DNA-PK [28]. However, DNA-PK signaling does not regulate MVM replication [14]. Instead, the MVM life cycle is dependent on signaling by the ATM kinase pathway [13,14,26]. Perhaps unsurprisingly for a virus that generates single-stranded DNA as it replicates, MVM replication inactivates signaling by the ATR kinase pathway that normally responds to single-stranded DNA breaks [29]. Based on these findings, we have previously proposed that ATR inactivation is a mechanism for MVM pathogenesis, which has evolved to enable MVM to inactivate the cellular signals that sense the single-stranded DNA virus genomes that may inhibit the viral life cycle [29,30]. 

In this study, we establish NS1 as a proxy marker for cellular DNA damage during MVM infection. Using this marker, we elucidate the cellular DDR pathways that modulate how MVM localizes to cellular sites of DNA damage. Strikingly, we have discovered that MVM localization to sites of cellular damage and early replication is dependent on signaling by the ATR kinase pathway, destined to be inactivated by MVM, and not the ATM or DNA-PK pathways. Our findings suggest that the single-stranded break repair processes aid in the establishment of MVM replication at the onset of infection.

## 2. Materials and Methods

### 2.1. Cell Lines and Virus, Viral Infections

Male murine A9 and female human U2OS cells were propagated in 10 percent Serum Plus (Sigma Aldrich) containing DMEM media (Gibco) supplemented with Gentamicin at 37 degrees Celsius and 5 percent carbon dioxide. A9 and U2OS cells were used as representative model systems for these studies because MVMp infects mouse cells and transformed human cells, respectively. Cell lines are routinely authenticated for mycoplasma contamination, and background levels of DNA damage are monitored by γH2AX staining. As A9 cells have smaller nuclei and higher background γH2AX levels relative to U2OS, imaging-based studies of NS1 localization to cellular DDR sites can be difficult to discern microscopically using this system. Therefore, U2OS cells were predominantly used for the laser microirradiation followed by imaging experiments. Wild-type MVMp virus was produced as previously described [26] and genome copies were quantified by Southern blotting [31]. MVMp infection was carried out at a Multiplicity of Infection (MOI) of 25 in all imaging assays. MVMp infection for Western blot analysis was carried out at an MOI of 10. The U2OS and A9 cells lines were obtained from Dr. David Pintel and have been previously published [26,27]. The CMV-NS1 plasmids were obtained from Dr. David Pintel and have also been previously published [14,26,27].

### 2.2. Cell Synchronization, DNA Damage, and Drug Treatments

A9 cells were parasynchronized in G0 phase of the cell cycle by isoleucine deprivation for 42 h as previously described [32,33]. Cells were infected with MVM upon release into complete DMEM medium (described above) for 16 h for imaging and 24 h for Western analysis. For Western blots, cells were pulsed with ATR inhibitor for 2 h (Berzosertib, Selleckchem S7102) starting at 12 h post-infection. The cells enter S phase approximately 12 h post-release [16]. For immunofluorescence, U2OS cells were infected with MVMp virus or transfected with NS1 expression plasmid for 16 h and treated with inhibitors 30 min prior to laser micro-irradiation unless otherwise indicated. The chemical inhibitors of DNA damage pathways were validated in U2OS cells that had been irradiated with 100 J/m^2^ of UV irradiation on a Stratagene crosslinker. Cells were allowed to recover for 30 min in the presence of the indicated DDR inhibitors (Table 1) before being processed for the phospho-specific epitopes described in Table 2. The number of respective DDR foci per nucleus were counted and plotted and are presented in Appendix A. 

### 2.3. Plasmids and Transfections

The NS1 open reading frame containing a mutation in the splice site for NS2 was cloned into the pcDNA3.1 mammalian expression vector as previously described [14,27]. Plasmids were transfected into U2OS cells for at least 16 h using the LipoD293 transfection reagent (SignaGen Laboratories, Frederick, MD, USA). 

### 2.4. Laser Micro-Irradiation for Immunofluorescence

Laser micro-irradiation was performed on 1 million U2OS cells cultured on glass-bottomed dishes (MatTek Corp, Ashland, MA, USA) infected with MVMp at an MOI of 25 or transfected with 1 µg of NS1-expression vectors for 16–20 h. Cells were sensitized with 3.5 μL of Hoechst dye (ThermoFisher Scientific, Madison, WI, USA) 5 min prior to micro-irradiation. Samples were irradiated using a Leica Stellaris DMI8 confocal microscope using 63× oil objective and 2× digital zoom with a 405 nm laser using 25 percent power at 10 Hz frequency for 2 frames per field of view. Regions of interest (ROIs) were selected across the nucleus (edges of the nuclei were demarcated and visualized by Hoechst staining). Samples were processed for immunofluorescence imaging and analysis (described below) immediately after micro-irradiation.

### 2.5. Immunofluorescence Assays

APAR body imaging was performed on 500,000 U2OS cells plated on glass cover slips and processed as described below. U2OS cells were pre-extracted with CSK Buffer (10 mM PIPES pH 6.8, 100 mM Sodium Chloride, 300 mM Sucrose, 1 mM EGTA, 1 mM Magnesium Chloride) and CSK with 0.5% Triton X-100 for 3 min each before fixation with 4% paraformaldehyde for 10 min at room temperature. Cells were washed in PBS before being permeabilized with 0.5% Triton X-100 in PBS for 10 min at room temperature. Samples were blocked with 3% BSA in PBS for 20 min, incubated with the primary antibody diluted in 3% BSA solution for 20 min, washed in PBS, and incubated with secondary antibody (tagged with the appropriate Alexa-Fluor conjugated fluorophores) for 20 min. Samples were washed in PBS and mounted on glass-bottomed dishes or glass slides with DAPI Fluoromount (Southern Biotech, Birmingham, AL, USA). Images were taken and processed on Leica Stellaris DMI8 Confocal microscope with 63× oil objective lens.

### 2.6. Image Analysis

Extent of relocalization of NS1 to micro-irradiation sites was quantified by measuring the intensity of NS1 signal along the γH2AX-labeled stripe. Following microirradiation, images were taken and processed on Leica Stellaris DMI8 Confocal microscope with 63× oil objective lens at 1.5× digital zoom. For cells that were NS1-positive by immunofluorescence, the intensity of NS1 signal over the laser micro-irradiated region was quantified using the plot profile tool on FIJI software. Signal intensities were measured at defined intervals from the left end to the right end of the region of interest (ROI) of irradiated nuclei. Values were averaged for 20–40 nuclei in 2–3 biological replicates at each position along the ROI. Quantification of NS1 relocalized to microirradiated sites relative to total NS1 levels in each individual cell was additionally measured by the ratio (NS1 intensity along the microirradiated stripe)/(total NS1 intensity in the nucleus). Images were processed with FIJI and an ROI outlined the nucleus using DAPI channel to measure total intensity of NS1 while another ROI was drawn outlining the microirradiated stripe using the gH2AX channel. 

### 2.7. Cell Cycle Analysis

Cell cycle analysis was performed by staining the total cellular DNA content in A9 fibroblasts with Propidium Iodide stain (Sigma Aldrich, Saint Louis, MO, USA). A9 cells were harvested at the indicated timepoints, washed in 1 mL of PBS, resuspended in 300 μL of PBS, fixed in 700 μL of chilled 100% ethenol overnight at 4 degrees Celsius. Cells were counted, samples were resuspended in 300 μL of PBS, treated with RNAse for 1 h at 37 degrees Celsius, and incubated with Propidium iodide normalized to cell number overnight. Cells were analyzed on a BD LSR Fortessa (University of Wisconsin-Flow Cytometry Laboratory, Madison, WI, USA) on the FL2 channel and assessed for cells in G0/G1, S, and G2/M phases of the cell cycle. 

### 2.8. MVMp Genome Replication Analysis

Cells were harvested at the indicated timepoints, pelleted and resuspended in Cell Lysis Buffer (2% SDS, 0.15 M Sodium chloride, 10 mM Tris pH 8, 1 mM EDTA). Lysed whole-cell extracts were proteinase K (NEB) treated overnight at 37 degrees Celsius. The genomic DNA was sheared with 25 G × 5/8 inch 1 mL needed syringe (BD Biosciences, Franklin Lakes, NJ, USA). The genomic DNA was purified using phenol:chloroform:isoamyl alcohol and precipitated in isopropanol. The DNA was washed in 70% ethanol and resuspended in 100 μL of TE buffer. MVMp genome replication was measured by Taqman-qPCR using forward and reverse primers (F: agccgctgaacttggactaa, R: ctccttggtcaaggctgttc) as well as a Taqman probe that was complementary to the plus strand of the viral genome (ccaaccatcccttaaaccct). 

### 2.9. Salt-Wash Chromatin Immunoprecipitation Combined with Quantitative PCR (swChIP-qPCR)

Salt-wash ChIP-qPCR assays were performed in nuclear extracts from MVMp-infected A9 cells as previously published [34,35]. Briefly, cells were washed with phosphate-buffered saline (PBS) before they were collected with HBE buffer (10 mM HEPES, 5 mM KCl, 1 mM EDTA) into Eppendorf tubes, centrifuged at 1000× *g* for 3 min at room temperature, aspirated, and resuspended in 500 μL HBE. Cells were lysed on ice for 10 min by the addition of 1% NP-40 (to a final concentration of 0.1%); sodium chloride (NaCl) was used to isolate the viral nucleoprotein complexes and crosslinked in 0.1% Formaldehyde for 10 min at room temperature. The crosslinking reactions were quenched in 0.125 M glycine, MVMp-nucleoprotein complexes purified using 3K Amicon Centrifugal Filter Units (Millipore), and purified into PBS and sonicated using a Diagenode Bioruptor Pico for 60 cycles (30 s on and 30 s off per cycle). The samples were incubated overnight at 4 degrees Celsius with the antibodies bound to Protein A Dynabeads (Invitrogen). Samples were washed for 3 min each at 4 degrees Celsius with low salt wash (0.01% SDS, 1% Triton X-100, 2 mM EDTA, 20 mM Tris-HCl pH 8, 150 mM NaCl), high salt wash (0.01% SDS, 1% Triton X-100, 2 mM EDTA 20 mM Tris-HCl pH 8, 500 mM NaCl), lithium chloride wash (0.25 M LiCl, 1% NP40, 1% DOC, 1 mM EDTA, 10 mM Tris HCl pH 8), and twice with TE buffer. Protein-DNA conjugates were eluted with SDS elution buffer (1% SDS, 0.1 M sodium bicarbonate), crosslinks were reversed using 0.2 M NaCl and Proteinase K (NEB) and incubated at 56 degrees Celsius overnight. DNA was purified using PCR Purification Kit (Qiagen) and eluted in 100 μL of Buffer EB (Qiagen). qPCR assays were performed with primers that were complementary to the MVM P4 and P38 promoters that have been previously published [35]. The respective P4 primer sequences were F: tgataagcggttcagggagt and R: ccagccatggttagttggtt, and P38 primer sequences were F: ccgaaaagtacgcctctcag and R: ccgcaacaggagtatttggt. 

### 2.10. Statistical Analysis

Statistical analysis of the imaging studies was performed using Graphpad Prism software. For ChIP-qPCR assays, the statistical significance of ATR association with MVM genome relative to that of IgG pulldown was determined by unpaired Student’s *t* test, with statistical significance denoted by ***, *p* < 0.001. For inhibitor treatments, the statistical significance of the number of foci was calculated using unpaired Student’s *t*-test with statistical significance denoted by ****, *p* < 0.0001. Statistical significance of MVM replication upon iATR treatment was assessed by unpaired Student’s *t*-test with statistical significance denoted by *, *p* < 0.05. For laser micro-irradiation assays, the statistical significance of the difference between NS1 relocalization between different conditions was calculated using one-way ANOVA, multiple comparisons test as previously described [27]. Statistical significance is designated by ****, *p* < 0.0001.

### 2.11. Western Blot Analysis

Cells were collected at specified time point and pellets were lysed on ice for 15 min in complete Radio-Immuno-Precipitation Assay (RIPA) buffer (20 mM Tris HCL pH 7.5, 150 mM NaCL, 10% glycerol, 1% NP-40, 1% sodium deoxycholate, 0.1% SDS, 1 mM EDTA, 10 mM trisodium pyrophosphate, 20 mM sodium fluoride, 2 mM sodium orthovanadate and 1× protease inhibitor cocktail (MedChemExpress)). Cell lysate was collected following centrifugation for 15 min at 17,000× *g* at 4 degrees Celsius. Protein sample concentration was calculated using BCA assay (Bio-Rad). A 5× protein loading dye (1 M Tris-HCl pH 8, 30% Glycerol, 100 mM DTT, 10 mM EDTA, 350 mM SDS, Bromophenol blue) was added to samples and boiled at 95 °C for 5 min. 

Equivalent protein levels were loaded onto a 10% SDS-PAGE gel for electrophoresis and subsequently transferred to a nitrocellulose membrane. Membranes were blocked for 30 min in 5% milk in TBS (20 mM Tris, 150 mM NaCl, pH 7.6) with 0.5% Tween-20. Membranes were incubated with the primary antibody in 5% milk in TBST for 1 h followed by three 5 min washes in TBST prior to incubation with HRP conjugated secondary antibody and ladder conjugate (Bio-Rad) in 5% milk in TBST for 30 min. All incubations were performed at room temperature. The membrane was incubated for 5 min with Clarity Western ECL Blotting Substrate (Bio-Rad) before imaging using a Li-COR-Fortessa scanner and analysis with Image Studio Lite software.

### 2.12. Antibodies

Primary antibodies were used in immunofluorescence (IF) and Western blot analysis (WB).

The mouse anti-NS1 monoclonal antibody (clone 2C9b) has been previously generated and extensively validated [14,26,27].

## 3. Results

### 3.1. NS1 Is a Proxy Marker for Cellular DNA Damage during MVM Infection

Upon infecting host cells, MVM genomes and non-structural proteins localize to pre-existing as well as induced cellular sites of DNA damage [26,27]. This localization is driven by NS1 associating with ACCA motifs on the MVM genome [27]. However, NS1 on its own activates a cytotoxic program in host cells. Therefore, to determine whether activation of this cytotoxic program by ectopic NS1 perturbs the host’s ability to recognize cellular DNA breaks, we expressed NS1 by transient transfection and monitored whether host cells recognize induced cellular DNA breaks using laser microirradiation followed by immunofluorescence. Using relocalized NS1 as a marker for induced DNA damage, we monitored the ability of cellular DDR proteins to recognize micro-irradiated sites. As shown in Figure 1A, NS1 relocalized to γH2AX sites, consistent with previously observed findings (Figure 1A, compare panel 1 to 2 [27]). This relocalization was also evident when monitored by staining for phosphorylated NBS1 (Figure 1A, panel 3), phosphorylated MDC1 (Figure 1A, panel 4), and phosphorylated RPA (Figure 1A, panel 5). Therefore, we have extended our observations of NS1 localization during ectopic expression to additional host DDR pathway proteins, finding NS1 relocalized to DDR sites marked by phosphorylated NBS1, MDC1, and RPA. Importantly, this interaction of NS1 with host DDR recognition markers does not impact their ability to recognize additional cellular DNA breaks.

While ectopic NS1 did not perturb the cellular detection of DNA damage, MVM replication is known to utilize ATM kinase signaling and inactivate the ATR kinase pathways [14,29]. To determine whether the actively replicating MVM genome inhibits the recognition of cellular DNA breaks, we analyzed the localization of cellular DDR proteins to laser micro-irradiated stripes during viral infection using NS1 as the DDR marker. As shown in Figure 1B, laser microirradiation of U2OS cells during MVM infection led to relocalization of NS1 to the induced cellular DDR sites marked by γH2AX, consistent with previously published findings (Figure 1B, compare panel 1 to 2 [27]). Using relocalized NS1 as a marker for induced DNA damage, we monitored the ability of cellular DDR proteins to recognize these sites. As shown in Figure 1A, DDR sites monitored by NS1 staining colocalized with DNA break recognition markers such as phosphorylated NBS1 (Figure 1B, panel 3), phosphorylated MDC1 (Figure 1B, panel 4), and phosphorylated RPA (Figure 1B, panel 5). These findings showed that MVM-infected cells still retain the ability to recognize additional cellular DNA breaks.

MVM infection leads to the inactivation of the ATR kinase pathway by inhibiting the single-stranded break transducer TopBP1 [29]. To confirm that NS1 is an alternative marker for cellular DDR during all stages of MVM infection, we performed laser microirradiation followed by immunofluorescence for NS1 and phosphorylated CHK1, which is activated downstream of ATR/TopBP1 [29]. Ectopically expressed NS1 robustly relocalized with cellular DDR sites that are marked by phospho CHK1 (Figure 1C, panel 1). Similarly, during early infection, when phosphorylation of CHK1 is still possible, NS1 colocalized with phospho CHK1 to mark cellular DDR sites (Figure 1C, panel 2). However, during late stages of infection, which is monitored by high nuclear levels of NS1 associated with inhibition of CHK1 [15], NS1 is still able to relocalize to the cellular DDR sites (Figure 1C, panel 3). These findings indicated that NS1 is a marker for cellular DNA breaks throughout all stages of MVM infection. 

### 3.2. NS1 Localization to Cellular DDR Sites Is Independent of ATM and DNA-PK Signaling but Depends on ATR Signaling

To systematically determine which cellular DDR signals drive the relocalization of NS1 to cellular sites of DNA damage, we used well-characterized DDR inhibitors in U2OS cells transfected with an NS1 expression vector combined with laser micro-irradiation assays ([27]; inhibitors validated in Appendix A). We investigated proteins in the ATR, ATM, and DNA-PK DDR pathways to determine which signals drive NS1 relocalization (Figure 2A,B). Inhibition of PARP polymerase that signals the initial activation of cellular DDR signals and local chromatin remodeling [36] did not impact NS1 relocalization (Figure 2C). Similarly, NS1 still relocalized to sites of DDR under inhibition of the single-strand DNA binding protein RPA (Figure 2D). However, in the presence of the pan ATM and ATR inhibitor caffeine, NS1 did not relocalize to sites of DDR (Figure 2H, top left), remaining as diffuse foci around the nucleus. To distinguish if this phenotype was driven distinctly by the ATM or ATR pathways, we used specific small-molecule inhibitors. While inhibition of ATM had no impact on NS1 relocalization (Figure 2H, top right), inhibition of ATR consistently resulted in a loss of relocalization (Figure 2H, lower left, grayscale images in Appendix A). This suggested that the ATR signaling pathway is involved in the relocalization of NS1 to sites of DDR. This decrease in NS1 localization relative to γH2AX in the presence of caffeine and iATR was consistently observed in more than 20 nuclei, and was statistically significant (Figure 2H, lower right). These findings were further corroborated by measuring the ratio of NS1 intensity over the microirradiated region to that of the pan-nuclear NS1 that revealed that NS1 relocalization to induced DNA break sites is attenuated in the presence of caffeine and iATR (Appendix A). Interestingly, the inhibition of the downstream ATR effector CHK1 did not impact the re-localization of NS1 to the laser micro-irradiated DDR site (Figure 2F). 

Surprisingly, although NS1 has been previously shown to interact with the cellular protein kinase CK2 [37] and CK2 interacts with MDC1 to transduce the signals to γH2AX [13,38], inhibition of CK2 activity did not attenuate the relocalization of NS1 to laser-microirradiated γH2AX sites (Figure 2E). Inhibition of DNA-PK signaling did not impact the ability of NS1 to localize to laser micro-irradiated sites (Figure 2G), suggesting NS1 relocalization discriminates between the cellular DDR PI3-kinase-like kinases [17].

### 3.3. ATR Associates with the MVM Genome and ATR Signaling Regulates the Early Replication of MVM

Since it has previously been reported that ATR colocalizes with MVM-NS1 in APAR bodies [29], and we observed that ATR inhibition during NS1 expression attenuated ectopic NS1 localization to cellular DDR sites, we hypothesized that ATR signaling may drive MVM localization during the early stages of infection. To determine whether ATR molecules associate with the MVM genome during infection, we performed chromatin immunoprecipitation assays on nuclear extracts containing the viral genomes that we have recently developed to remove the secondary effects of host-associated cellular DDR sites [34,35]. ChIP-qPCR assays of ATR compared with NS1 pulldowns revealed that ATR molecules associated strongly with the MVM genome at 16 hpi at the P4 (Figure 3A) and P38 promoters (Figure 3B). The pulldown efficiencies of ATR were equivalent to that of NS1, which is known to bind covalently to the 5’ end of the MVM genome and associate with ACCAACCA consensus motifs [3,4,39,40]. To examine whether ATR signaling impacted NS1 localization to cellular sites of DNA damage, we pulsed MVM infected U2OS cells with ATR inhibitor for 30 min prior to laser microirradiation. As shown in Figure 3C, in the presence of ATR inhibitor, MVM-produced NS1 relocalization to induced DDR sites was attenuated at higher concentrations of iATR (grayscale images in Appendix A). These assays qualitatively suggested that ATR signaling is required for initial localization of NS1 to cellular DDR sites during infection. To determine how ATR signaling regulates MVM life cycle, we treated A9 cells that were synchronized by Isoleucine deprivation and pulsed with ATR inhibitor at 12 h post-release as shown in the schematic (Figure 3D). This strategy of ATR inhibition did not impact A9 cell entry into S phase (which occurs at approximately 12 h post-release into complete media) as determined by cell cycle analysis at 14 hpi (Figure 3E, compare 4 columns on the right). Under these synchronization conditions, in the presence of the ATR inhibitor, MVM-APAR bodies were smaller and formed foci that showed a distinct separation between NS1 and γH2AX (Figure 3F and profiles on the right). To determine the impact of ATR inhibition on MVM replication, the A9 cells were synchronized and pulsed with iATR at 12 hpi, washed at 14 hpi, and harvested at 24 hpi, as schematized in Figure 3G. This led to a decrease in MVM replication, as measured by NS1 levels (Figure 3H, lanes 3 and 4), which also correlated with a decrease in replicating viral genomes (Figure 3I). Taken together, these findings confirmed that inhibition of the ATR signaling pathway attenuates the ability of MVM-NS1 to localize to cellular sites of DNA damage, thereby regulating how viral replication centers are established. 

## 4. Discussion

In this study, we have validated that NS1 is a bona fide marker for parvovirus-induced cellular DNA damage. Additionally, we have systematically perturbed the major cellular DDR signaling pathways, determining that ATM, PARP, and DNA-PK are not involved in regulating NS1 localization to induced cellular DDR sites. Strikingly, ATR inhibition decreases NS1 relocalization to cellular DDR sites, leading to attenuated MVM replication. Taken together, these findings suggest that although ATR signaling is inactivated by MVM infection at late stages, it is required at the early stages of the viral infection to initially establish APAR bodies. 

Mass spectrometry studies investigating the host proteins that associate with MVM NS1 have previously discovered that NS1 interacts with the host protein Protein Kinase CK2 (CK2, [41]). In this regard, the NS1 protein of autonomous parvoviruses functions as an adaptor, connecting CK2 with cytoskeletal proteins such as tropomyosin, to cause cytopathic effects in permissive cells [37]. Additionally, since CK2 also has an established role in connecting MDC1 and NBS1 [42], we initially hypothesized that CK2 might serve as a connecting link between NS1 and cellular DDR proteins. However, the CK2 inhibitor did not impact the relocalization of MVM-NS1 to cellular DDR sites. Interestingly, CK2 is made up of two subunits: a catalytic alpha subunit, which interacts with NS1 during MVM infection, and a regulatory beta subunit. While the inhibitor used in this study targeted the function of the alpha subunit, it remains possible that the catalytic function is not required for NS1-CK2α interaction. These findings suggest that the role of CK2’s catalytic activity in regulating the MVM life cycle is independent of the localization of NS1 (and by extension MVM) to cellular DDR sites. 

Much of our understanding of virus–host interactions comes from transfection of viral proteins ectopically. However, our findings in this study suggest there are phenotypic differences in how viral proteins interact with the host during infection versus ectopic expression. This may be due to the absence of the replicating viral genomes. Alternatively, it is possible that active virus replication causes global changes that modulate NS1-DDR interactions differently. While the inhibition of ATR obliterates NS1 relocalization to sites of DNA damage in overexpression, during infection the presence of increasing MVM DNA molecules and additional viral protein expression presents the potential for redundant pathways to be activated, which can then induce relocalization. This may be contributing to why relocalization is inhibited in cells at early stages of infection but not late stages. One limitation of current laser micro-irradiation studies is the inability to perform synchronous infection assays with U2OS cells, which is the established model system for studying DDR signaling. This makes it difficult to rigorously evaluate how NS1 localization to cellular micro-irradiated sites changes over the course of infection. Time lapse live cell imaging experiments and laser micro-irradiation studies in synchronous MVM infection systems will provide more clues on these aspects of the viral life cycle. 

While relocalization was only attenuated during early infection by inhibition of ATR, APAR body formation was still impacted and correlated with decreased overall virus replication. Previous work investigating APAR body formation has shown that while at lower resolutions NS1 appears to be directly associated with γH2AX, at higher resolution, MVM APAR bodies form in distinct pockets surrounded by or directly next to γH2AX [13,14,26]. In the presence of the ATR inhibitor, there is a greater separation of NS1 from that of cellular γH2AX. One possible interpretation for this observation is that MVM localizes inefficiently to cellular sites of DNA damage in the absence of ATR signals. This builds on our observations that NS1-DDR interactions are dependent on ATR signaling. Our observations with the ATR inhibitor also suggest that while NS1 initially associates with γH2AX to form APAR bodies, the inability of NS1 to relocalize and expand to new cellular DDR sites results in smaller APAR bodies. This is further supported by decreased levels of NS1 and viral genome later in infection in the presence of the ATR inhibitor. In stark contrast, the virus-induced cellular DNA damage signals activate the ATM kinase pathways [14]. We have recently discovered that efficient MVM expression and replication depend on MRE11 in an MRN-complex-independent manner [35]. These findings suggest that MVM utilizes distinct aspects of cellular DDR pathways at different stages of the viral life cycle, establishing a distinction between the signals necessary for expression from those required for localization. 

An attractive mechanism by which PIKKs may regulate NS1 localization to cellular DDR sites is via phosphorylation of their SQ/TQ motifs. Cellular Serine/Threonine residues that are associated with glutamine residues on the primary amino acid sequence are phosphorylation targets of ATM and ATR kinases [43]. Further supporting this assertion, the Rep 68/78 protein of AAV2 localizes to cellular DDR sites in an ATM/ATR-dependent manner [44]. In line with these expectations, the MVM NS1 protein contains two SQ and two TQ motifs. We hypothesize that these are the residues that serve as the phosphorylation substrates for ATM and ATR kinases. Our ChIP studies show that ATR associates with the MVM genome. However, based on our findings, it remains unknown why these sites might be specifically targeted by ATR and not ATM. Future studies will focus on dissecting these conundrums on parvovirus–host genome interactions. 

Prior studies on MVM pathogenesis established that MVM infection inactivates CHK1 (downstream of ATR) by inhibiting the activation of the transducer TopBP1 [29]. However, our findings suggest that while ATR may be inactivated at late stages, it is required at the early stages of infection to establish viral replication centers in the nuclear compartment. Perhaps ATR interaction at early stages facilitates the opening of the viral genome for access by host replication proteins, whereas at late stages this enhances genome instability and chromosome fragmentation. Alternatively, it is possible that ATR inactivation at the late stages of infection is a mechanism evolved by parvoviruses to evade integration into the host genome. It remains unclear, however, whether this requirement for ATR pathway inactivation is induced on the viral genome or on the host genome. Interestingly, Adenovirus infection induces distinct virus and host-mediated ATM signals [45]. We speculate that MVM, by virtue of being a single-stranded DNA virus, interacts with the host ATR pathway via a similar route. Future studies will elucidate how distinct single-stranded DNA repair pathways on the viral versus cellular genomes drive viral pathogenesis and can be used for oncolytic virotherapies. 

## Figures and Tables

**Figure 1 viruses-15-01243-f001:**
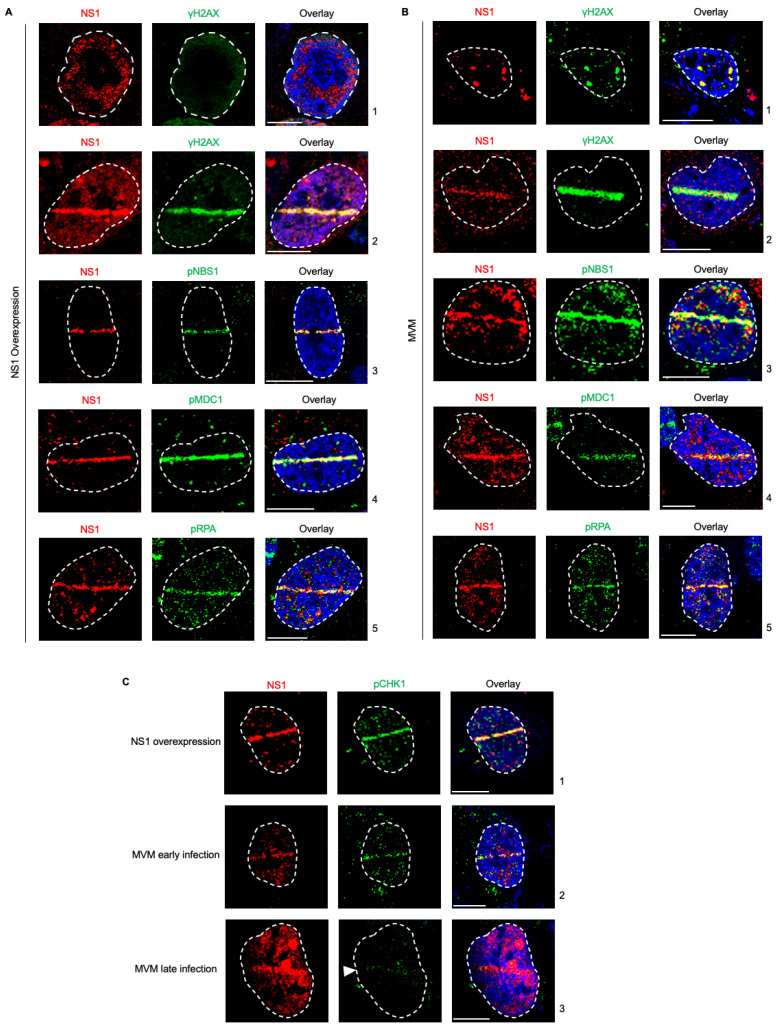
NS1 is a proxy marker for cellular DNA damage during MVM infection. (**A**) U2OS cells transfected with NS1-expressing vectors for 16 h were damaged across the nucleus using laser micro-irradiation and monitored by NS1-DDR staining as indicated. (**B**) U2OS cells infected with MVM at an MOI of 25 for 16 h were damaged across the nucleus using laser micro-irradiation and monitored by NS1-DDR staining as indicated. (**C**) Relocalization of NS1 produced during different stages of MVM infection or ectopic expression were monitored by relocalization to DDR sites marked by phosphorylated CHK1. Viral non-structural protein NS1 (red), respective DDR markers (green), and pan-nuclear content were visualized by DAPI staining (blue). White dashed lines demarcate the nuclear boundaries, white bars represent 10 μm, and white arrowhead in (**C**) panel 3 indicates the location of the laser micro-irradiated stripe. The ratios of NS1 along the laser stripe to that of total nuclear NS1 in 1C are as follows: panel 1 (NS1): 0.404; panel 2 (MVM early): 0.293; and panel 3 (MVM late): 0.197.

**Figure 2 viruses-15-01243-f002:**
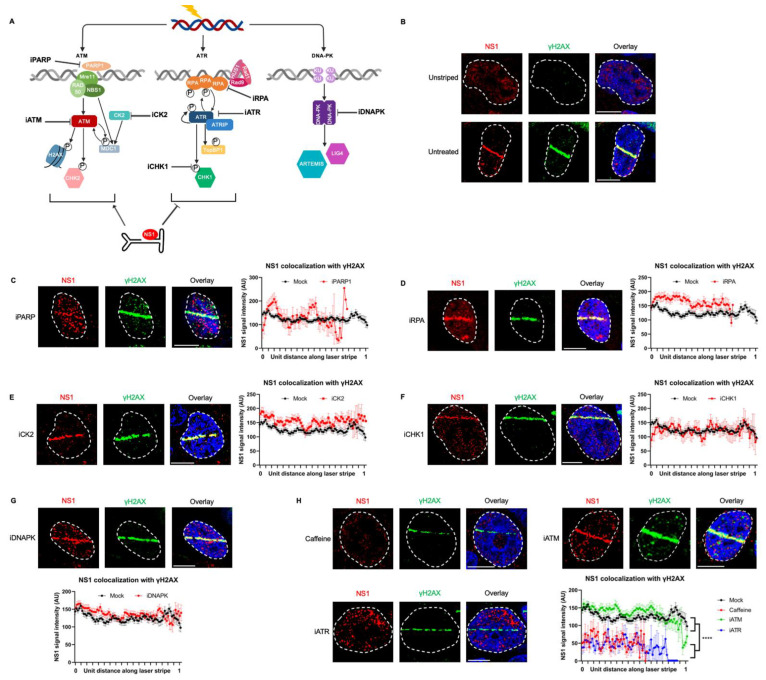
NS1 localization to cellular DDR sites is independent of ATM and DNA-PK signaling but depends on ATR signaling. (**A**) Schematic of cellular DNA damage response mediated by PI3-Kinase-like-Kinases with the relevant proteins inhibited in part (**B**–**H**) indicated as shown. (**B**–**H**) U2OS cells were transfected with NS1 overexpression vector for 16 h, treated with the indicated DDR inhibitors 30 min prior to laser micro-irradiation, and processed for NS1-γH2AX colocalization. Viral non-structural protein is depicted by NS1 (red), cellular DDR by γH2AX (green), and pan-nuclear content was visualized by DAPI staining (blue). White dashed lines demarcate the nuclear boundaries and white bars represent 10 μm. The concentrations of the inhibitors used are described in Materials and Methods. Representative relocalization upon laser micro-irradiation of U2OS cells expressing NS1 was compared with relocalization upon (**C**) PARP inhibition, (**D**) RPA inhibition, (**E**) CK2 inhibition, (**F**) CHK1 inhibition, (**G**) DNAPK inhibition, and (**H**) inhibition of the ATM/ATR pathway by caffeine or specific ATM/ATR inhibitors. The average intensity of NS1 localizing to γH2AX-labelled nuclear sites between mock-treated cells and inhibitor-treated cells was averaged over at least 20 independent nuclei, represented in the adjoining profiles with the error bars representing SEM.

**Figure 3 viruses-15-01243-f003:**
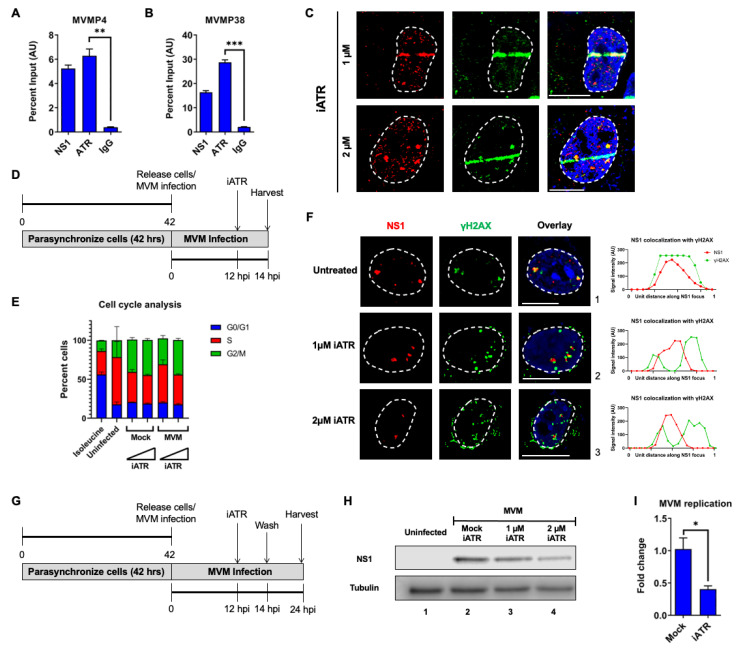
ATR associates with the MVM genome and ATR signaling regulates the early replication of MVM. Parasynchronized A9 cells were infected with MVM at an MOI of 10 for 16 hpi before being processed for salt-wash ChIP-qPCR assays on the viral nucleoprotein extracts for their association with NS1, ATR, and IgG as negative control. The pulldowns were assessed for (**A**) P4 and (**B**) P38 sequences using qPCR analysis. (**C**) U2OS cells infected with MVM for 16 hpi were treated with iATR at concentrations of 1 µM (top panel) and 2 µM (bottom panel) for 30 min prior to laser micro-irradiation before being processed for NS1-γH2AX colocalization. (**D**) Schematic of MVM synchronization for cell cycle analysis and APAR body imaging. A9 cells were synchronized by isoleucine deprivation for 36–42 h (as shown in Figure 3D) before being released into complete DMEM media and infected with MVM at an MOI of 10. At 12 hpi, the cells were pulsed with 1 µM and 2 µM concentrations of iATR for 2 h and harvested at 14 hpi for (**E**) cell cycle analysis and (**F**) APAR body imaging. Viral non-structural protein is depicted by NS1 (red), cellular DDR by γH2AX (green), and pan-nuclear content was visualized by DAPI staining (blue). White dashed lines demarcate the nuclear boundaries and white bars represent 10 μm. On the right-hand panels of the respective images in (**F**), the profiles of NS1 (red) and γH2AX staining along the transverse section of the APAR body are shown. (**G**) Schematic of MVM infection for evaluating MVM replication. (**H**) Synchronized A9 cells were infected with MVM for 24 h. Cells were treated with iATR at the indicated concentrations from 12 hpi to 14 hpi before being washed out. Virus replication was evaluated using Western blots for NS1 levels compared with Tubulin as loading control. (**I**) MVM replication at 24 hpi was evaluated in the absence (mock) or presence of iATR at 2 µM using Taqman qPCR assay directed against the plus strand. Statistical analysis was performed by paired Student’s *t*-test with *p* values represented by: * *p* < 0.05, ** *p* < 0.01 and *** *p* < 0.001.

**Table 1 viruses-15-01243-t001:** Summary of inhibitors used.

Inhibitor	Target	Supplier	Catalog Number	Dosage
Olaparib	PARP	Selleckchem	S1060	1 µM
TDRL-505	RPA binding	Millipore Sigma	5.30535	50 µM
Caffeine	ATM and ATR	Millipore Sigma	W222402	2.5 µM
KU-55933	ATM	Selleckchem	S1092	7 µM
Berzosertib	ATR	Selleckchem	S7102	1 µM and 2 µM
NU7441 (KU-57788)	DNA-PK	Selleckchem	S2638	10 µM
CAS 17374-26-4	CK2	Millipore Sigma	218697	10 µM
CHIR-124	CHK1	Selleckchem	S2683	10 µM

**Table 2 viruses-15-01243-t002:** Summary table of antibodies used.

Antibody	Supplier	Catalog Number	Application
Tubulin (Clone DM1A)	Sigma	05-829	WB
Mre11	Cell Signaling	4895	WB
RPA phospho-Ser4, Ser8	Thermo Fisher	A300-245A	IF
EXO1 phospho-Ser746	Sigma	ABE1066	IF
CHK1 phospho-Ser345	Cell Signaling	23415	IF
CK2 phospho-Ser/Thr	Cell Signaling	8738	IF
DNAPKcs phospho-Ser2056	Cell Signaling	68716	IF
Poly/Mono-ADP Ribose (E6F6A)	Cell Signaling	83732	IF
ATM phospho-Ser1981	Cell Signaling	13050	IF
ATR phospho-Ser428	Cell Signaling	2853	IF
NBS1 phospho-Ser95	Cell Signaling	3002	IF
MDC1 phospho-T4	Abcam	Ab35967	IF
anti-Mouse-AF568	Thermo Scientific	A11004	IF
anti-Rabbit-AF488	Thermo Scientific	A11034	IF
Anti-mouse IgG, HRP-linked	Cell Signaling	7076	WB
Anti-rabbit IgG, HRP-linked	Cell Signaling	7074	WB

## Data Availability

All relevant data supporting the findings of this study are available within the paper and its Appendix A. All other data are available from the corresponding author on request.

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
