# Peer review of "The Autonomous Parvovirus Minute Virus of Mice Localizes to Cellular Sites of DNA Damage Using ATR Signaling"

_viruses, 2023, doi:10.3390/v15061243_

Round 1

Reviewer 1 Report

Summary:

The paper discusses how Minute Virus of Mice (MVM) uses its NS1 protein to localize to DNA damage response sites, where it establishes viral replication centers. While activating the ATM kinase pathway, MVM inhibits the ATR kinase pathway. The researchers used chemical inhibitors to investigate how MVM targets DDR sites and found that NS1 localization depends on ATR signaling. Additionally, inhibiting ATR after the S-phase entry led to decreased MVM replication. This suggests that ATR signaling plays a crucial role in MVM's initial localization to DDR sites before being shut down by the virus's replication.

General comment:

In general, authors introduced a strategy that MVM used to localize to DNA damage sites and establish viral replication. The work is well supported, and the evidence is laid out quite clearly. A few points would need clarification and more data would need to be provided before it could be considered for publication in Viruses.

Specific comment:

For most of the claims in the paper, the authors would need to provide quantification to back their claim instead of only showing the representative images. For example, for the localizing effect of NS1 protein, authors would need to provide quantification of the percentage of the NS1 staining that overlap with the DNA damage response site markers. Same applies to figure 2 and figure 3.

For presentations, the authors should select images that has similar scale bar within a panel, the variation of the size of the cells and images are not ideal. Please add an extra explanation if the variation is inevitable. This also emphasized the need for quantification as mentioned above.

Author Response

For most of the claims in the paper, the authors would need to provide quantification to back their claim instead of only showing the representative images. For example, for the localizing effect of NS1 protein, authors would need to provide quantification of the percentage of the NS1 staining that overlap with the DNA damage response site markers. Same applies to figure 2 and figure 3.

Response: Thank you for the suggestion. We have provided rigorous quantifications of the histograms of NS1 colocalizing with cellular gamma H2AX at laser micro-irradiated stripes in Figure 2 in the presence of DDR inhibitors. This data has been generated from at least twenty nuclei of laser micro-irradiated cells treated with the respective inhibitors over three independent biological replicates. These findings have bolstered our overall conclusions. The laser striping assays in Figure 1 are a qualitative assessment of NS1 as a marker for DNA damage proteins. We have provided quantifications of NS1 associating with the cellular DDR sites in Figure 3F. (Page 10 and 12)   

For presentations, the authors should select images that has similar scale bar within a panel, the variation of the size of the cells and images are not ideal. Please add an extra explanation if the variation is inevitable. This also emphasized the need for quantification as mentioned above.

Response: Thank you for the suggestions. We have modified the images to have equivalently-sized scales throughout the manuscript (Page 8, 10 and 12).

Reviewer 2 Report

The current manuscript provides new information about the regulatory mechanisms of MVM to target replication centers to DNA damage response sites after infection. Using chemical inhibitors of candidate regulatory DDR proteins, this is achieved through co-localization studies of the parvoviral non-structural protein NS1 with candidate markers of DDR sites. This is indeed an interesting study providing new insights into the regulatory mechanisms to initiate parvovirus replication and in consequence cell permissiveness for propagation and spreading.

Although, the overall study appears convincing and is very well documented with the current literature, it could gain more credibility if the authors would indicate the average amount of cells examined for the individual experiments in Materials and Methods and the amount of co-localization in the Figures, if possible.

In addition, there are some minor points that should be addressed:

1)     NS1/CKIIalpha interaction leads to a change of substrate specificity as compared to the CKIIalpha/beta complex. Therefore, it would be of interest to know, how CAS17374-26-4 is functioning in order to inhibit kinase activity:  While inhibition of the complex between the regulatory subunit CKIIbeta with the catalytic subunit CKIIalpha would potentially have no impact on NS1/CKIIalpha, functioning, an inhibition of the catalytic site of CKIIalpha could effectively knock-out the functioning of both.

2)     Figure 2: A vs. B are switched regarding the figure legend.

3)     There is a misspelling in lane 8 of the abstract: NS1 “locallization”.

Author Response

Although, the overall study appears convincing and is very well documented with the current literature, it could gain more credibility if the authors would indicate the average amount of cells examined for the individual experiments in Materials and Methods and the amount of co-localization in the Figures, if possible.

Response: Thank you for the suggestion. We have provided rigorous quantifications of the histograms of NS1 colocalizing with cellular gamma H2AX at laser micro-irradiated stripes in Figures 2 in the presence of DDR inhibitors. This data has been generated from at least twenty nuclei of laser micro-irradiated cells treated with the respective inhibitors over three independent biological replicates, and have also been described in the Materials and Methods section of the revised manuscript. These findings have bolstered our overall conclusions (Page 4 and 5).   

In addition, there are some minor points that should be addressed:

  • NS1/CKIIalpha interaction leads to a change of substrate specificity as compared to the CKIIalpha/beta complex. Therefore, it would be of interest to know, how CAS17374-26-4 is functioning in order to inhibit kinase activity: While inhibition of the complex between the regulatory subunit CKIIbeta with the catalytic subunit CKIIalpha would potentially have no impact on NS1/CKIIalpha, functioning, an inhibition of the catalytic site of CKIIalpha could effectively knock-out the functioning of both.

Response: This is an important point. The CK2 inhibitor CAS17374-26-4 inhibits the function of CK2 alpha subunit. We hypothesize that inhibiting this activity likely attenuates the ability of the native CK2 complex to interact with NS1 and thereby is unable to bridge the viral proteins to cellular DDR pathways, namely MDC1 and NBS1. Future studies will dissect this aspect of NS1-CK2 interaction in DDR pathway localization using additional commercially available chemical inhibitors. We have clarified this point in the Discussion section of the manuscript to focus on the currently known impact of the inhibitor on the catalytic function of CK2 alpha subunit (Page 11, last paragraph).

2)     Figure 2: A vs. B are switched regarding the figure legend.

      Response: Thank you for the suggestions. We have switched the description of Fig. 2 A/B figure legends to keep the legends consistent with the figures. We have additionally made significant changes to Figure 2 to incorporate rigorous quantification of the laser micro-irradiation studies (Page 10).  

3)     There is a misspelling in lane 8 of the abstract: NS1 “locallization”.

      Response: Thank you for the identification of the typo. We have corrected the misspelling (Page 1, Abstract).

Reviewer 3 Report

In their manuscript entitled “The autonomous parvovirus Minute Virus of Mice localizes to cellular sites of DNA damage using ATR signaling”, Larsen and Majumder extend their previous findings on the relationship between MVM and DDR. They show that both during MVM infection and upon NS1 exogenous expression in the absence of additional viral proteins, NS1 localizes to sites of DNA repair after nuclear irradiation mediated by laser light. Interestingly the aforementioned NS1 ability is impaired in both systems after treatment with compounds interfering with the ART pathway but not with compounds interfering with the ATM or DNA-PK pathways. This is interesting, because the ATR pathway has been previously shown by the same authors to be inactivated by MVM at late times post infection, and therefore suggests a complex relationship between MVM and the DDR. Overall, the manuscript is well organized and clearly presents potentially interesting and novel, although preliminary findings. Unfortunately, there are several issues which need to be addressed, including the inclusion of several controls and additional experimental evidences, to support the authors’ conclusions. Furthermore, the manuscript apparently lacks any sort of image and statistical analysis, leaving some concerns regarding the soundness of the data shown. Overall, the manuscript appears too preliminary to warrant publication at this stage. Since I believed that all the issues could be easily addressed by the authors, I strongly recommend a they submit a revised version of their work. Please note that I do not believe that such revision could be performed in the usual 1-week time asked for major revisions by MDPI journals, and I therefore suggest rejection - and would encourage resubmission when the manuscript has been revised as suggested.

Major points.

1)    Pharmacological treatments lack control experiments showing specific inhibition of the targeted pathways, and absence of cytotoxic effects. This must necessarily be included. For example, it must be shown that at the specific dosages and conditions used, Berzosertib specifically inhibits the ATR pathway. This is particularly important in Figure 3, where two different concentrations are used, the lower not exerting any activity on NS1 localization. This can be easily addressed, for example by staining of appropriate markers/effectors of the different pathways. 

2)    It is essential to show the effect of early inhibition of the ATR pathway on MVM life cycle, including early and late gene expression, and viral genome replication, to further extend on the findings shown in Figure 3 c.

3)    The manuscript is based almost exclusively on CLSM images. Unfortunately, only one image is shown per each condition, often containing only one cell. Therefore, the number of replicates should be indicated, along with the number of analyzed cells per condition per replica, along with proper image analysis results to allow statistical analysis of the results. It would be important for example to quantity the proportion of NS1 (and other DDR markers) which relocalize to the DNA damage area at the single cell level. 

Minor (but important!) points

1)    All single channels from CLSM should be replaced with grayscale images which allow a much better understanding ox pixel intensity. 

2)    The manuscript is mainly based on CLSM images. Unfortunately, the Figures are often not easy to follow, due to the extensive used of numbering – whose meaning is often unclear. I suggest removing such numbering where not needed (for example from Figure 1 and 3).

3)    Scale bars are highly variable in size and sometimes completely absent (see Fig. 3B). This makes difficult to address nuclei shape and possible morphological alterations due to viral infection/drug treatment. I suggest using the same magnification for each micrograph if possible. 

4)    Arrowheads are often mentioned in Figure legends but absent from the micrographs.  

5)    Figure 1ab

a.     the amount of NS1 relocalized to sites of DDR appears very variable from cell to cell (and often a single cell is shown per condition). Quantification and statistical analysis is needed (see major points).  

6)    Figure 1c

a.     what is “early time” and “late time”? can the authors be more specific? 

b.     I’m not sure that at late times NS1 is still recognizing sites of DNA damage… another marker of DDR - alternative to pCHK1 and independent of ATR/TopBq should be shown

c.     NS1 staining in the “late time” infected cells is a bit strange and looks different from the previous panels. 

7)    Figure 2a

a.     I don’t see a point for number “5”

b.     “P” and “numbers” should be somehow differentiated graphically, otherwise the Figure is difficult to read. 

c.     The amount of NS1 relocalized to DDR sites in (3) seems much lower than under other conditions. This further highlights the need of appropriate quantitative image and statistical analysis.

d.     Which drugs are 6 and 7? What do they block exactly? I find this numbering system quite confusion; it might be better to use drug names instead of numbers? 

8)    Figure 3a

a.     Since MVM is known to inhibit ATR at late times after infection it would be important to repeat the experiment in conditions were infection is synchronized? 

b.     In panel 3 it looks like NS1 at late times “L” still efficiently localized to DDR sites upon caffeine treatment, the authors should comment on that. I did not find any mention to this in the text. 

c.     How Early and Late infected cells should be clearly described in the material and methods and results sections.  

9)    Materials and Methods

a.     The section “Image analysis” is completely missing. The same is true for the section “statistical analysis” 

10) Results

a.     “the cellular Casein Kinase II”. Casein Kinase II is a misnomer, and has not been used for decades. The correct name is “protein kinase CK2”.

b.    “However when treated with a higher caffeine.” I think the word “concentration” is missing here.

11) Discussion

a.     “These findings suggest that the role of CK2 in regulating MVM life cycle is independent of the localization of NS1 “. Please describe the role of CK2 in MVM life cycle since I am not sure all the readers are familiar with that. 

English is of very high quality (much better than mine)

Author Response

Pharmacological treatments lack control experiments showing specific inhibition of the targeted pathways, and absence of cytotoxic effects. This must necessarily be included. For example, it must be shown that at the specific dosages and conditions used, Berzosertib specifically inhibits the ATR pathway. This is particularly important in Figure 3, where two different concentrations are used, the lower not exerting any activity on NS1 localization. This can be easily addressed, for example by staining of appropriate markers/effectors of the different pathways.

Response: Thank you for the suggestion. We have performed extensive validation studies of the inhibitors used in this manuscript in U2OS cells that are undergoing UV-induced cellular DNA damage. These findings are presented in Supplementary figure S1. Additionally, we have simplified our treatment conditions with Caffeine to use only the lower concentration (Supplementary figure S1 and Page 12).   

2)    It is essential to show the effect of early inhibition of the ATR pathway on MVM life cycle, including early and late gene expression, and viral genome replication, to further extend on the findings shown in Figure 3 c.

Response: Thank you for the suggestion. We have performed additional experiments verifying the association of ATR with the MVM genome using ChIP assays in viral salt-wash extracts (Fig. 3A, 3B) Additionally, we have evaluated the impact of ATR inhibition on the MVM life cycle by evaluating protein production downstream of gene expression (Fig. 3H), quantitative viral genome replication (Fig. 3I), quantitative viral replication center formation (Fig. 3F), as well as host cell cycle entry (Fig. 3E). All of these findings are presented in modified Figure 3 (Page 12).    

3)    The manuscript is based almost exclusively on CLSM images. Unfortunately, only one image is shown per each condition, often containing only one cell. Therefore, the number of replicates should be indicated, along with the number of analyzed cells per condition per replica, along with proper image analysis results to allow statistical analysis of the results. It would be important for example to quantity the proportion of NS1 (and other DDR markers) which relocalize to the DNA damage area at the single cell level.

Response: Thank you for the suggestion. We have provided rigorous quantifications of the histograms of NS1 colocalizing with cellular gamma H2AX at laser micro-irradiated stripes in Figures 2 in the presence of DDR inhibitors. This data has been generated from at least twenty nuclei of laser micro-irradiated cells treated with the respective inhibitors over three independent biological replicates. These findings have bolstered our overall conclusions (Page 10). 

Minor (but important!) points

1)    All single channels from CLSM should be replaced with grayscale images which allow a much better understanding ox pixel intensity. 

Response: We have provided grayscale images of the important and relevant findings in Supplementary figure S2 to enhance the clarify of the findings and make them easy to follow. We have also provided extensive quantification of the inhibitor treatment studies along with statistical analysis to support our conclusion that ATR signaling regulates NS1 localization to cellular DDR sites (Supplementary figure S2 and Page 10).  

2)    The manuscript is mainly based on CLSM images. Unfortunately, the Figures are often not easy to follow, due to the extensive used of numbering – whose meaning is often unclear. I suggest removing such numbering where not needed (for example from Figure 1 and 3).

Response: We have modified the figures (particularly figures 2 and 3) to be easier to follow by removing references to subpanel numbers (Page 10 and Page 12).   

3)    Scale bars are highly variable in size and sometimes completely absent (see Fig. 3B). This makes difficult to address nuclei shape and possible morphological alterations due to viral infection/drug treatment. I suggest using the same magnification for each micrograph if possible.

Response: We have modified the relevant images to ensure the nuclear sizes are consistent across subpanels (Page 8, 10, 12).  

4)    Arrowheads are often mentioned in Figure legends but absent from the micrographs.

Response: We have provided clarifications and descriptions of arrowheads where applicable and removed them from panels where they were non-essential (Page 8).   

5)    Figure 1ab

  1. the amount of NS1 relocalized to sites of DDR appears very variable from cell to cell (and often a single cell is shown per condition). Quantification and statistical analysis is needed (see major points). 

Response: We have provided extensive quantification of all the images in figure 2. The goal of figure 1 was to qualitatively establish that other DDR proteins can be used as proxies for DDR markers. This knowledge is important because MVM infection has been shown to modulate distinct cellular DDR pathway proteins. Therefore, we have not provided statistical analysis of this figure.  

6)    Figure 1c

  1. what is “early time” and “late time”? can the authors be more specific? 
  2. I’m not sure that at late times NS1 is still recognizing sites of DNA damage… another marker of DDR - alternative to pCHK1 and independent of ATR/TopBq should be shown
  3. NS1 staining in the “late time” infected cells is a bit strange and looks different from the previous panels.

Response:  As we acknowledge in our discussion section, these studies with U2OS cells have a limitation in not being able to synchronize. Therefore, assessments of “early” versus “late” timepoints in MVM infection of U2OS cells are based on NS1 levels. We have also clarified this in Page 9, last paragraph of Section 3.1. 

7)    Figure 2a

  1. I don’t see a point for number “5”
  2. “P” and “numbers” should be somehow differentiated graphically, otherwise the Figure is difficult to read. 
  3. The amount of NS1 relocalized to DDR sites in (3) seems much lower than under other conditions. This further highlights the need of appropriate quantitative image and statistical analysis.
  4. Which drugs are 6 and 7? What do they block exactly? I find this numbering system quite confusion; it might be better to use drug names instead of numbers?

Response: We have revised this figure entirely to simplify and clarify the inhibitors that are used to target distinct parts of the DDR signalling pathways in Figure 2A. This includes removing the numbers in designating the subpanels. In order to quantify NS1 levels across multiple nuclei and how they relocalize to induced DDR regions, we have provided relocalization profiles averaged across multiple nuclei with error bars. We have provided the names of the specific drug targets in the text and are further described in the drug table in materials and methods. The previously described drug numbers 6 and 7 are respectively ATM inhibitor and ATR inhibitor. The ATM inhibitor inhibits phosphorylation of the downstream targets in the ATM kinase pathway. The ATR inhibitor Berzosertib prevents phosphorylation of the downstream substrates in the ATR kinase pathway (Page 10).   

8)    Figure 3a

  1. Since MVM is known to inhibit ATR at late times after infection it would be important to repeat the experiment in conditions were infection is synchronized? 
  2. In panel 3 it looks like NS1 at late times “L” still efficiently localized to DDR sites upon caffeine treatment, the authors should comment on that. I did not find any mention to this in the text. 
  3. How Early and Late infected cells should be clearly described in the material and methods and results sections.

Response: Thank you for the suggestion. We have assessed the impact of ATR inhibition on MVM replication during synchronous infection in A9 cells in the revamped Figure 3. We have also simplified the findings of the impact of ATR inhibition on NS1 relocalization during infection. We have described these in the appropriate section of the Results section and Discussed them further in discussion (Page 12).     

9)    Materials and Methods

  1. The section“Image analysis” is completely missing. The same is true for the section “statistical analysis”

Response: We have provided extensive descriptions of Image analysis (Page 4) and Statistical analysis (Page 5) in the Materials and Methods section. Thank you for pointing this out.

10) Results

  1. “the cellular Casein Kinase II”. Casein Kinase II is a misnomer, and has not been used for decades. The correct name is “protein kinase CK2”.
  2. “However when treated with a higher caffeine.” I think the word “concentration” is missing here.

Response: Thank you for the suggestion. We have modified the mention of Casein Kinase II to “protein kinase CK2” as suggested and removed the section on higher caffeine concentration to simplify the narrative of the manuscript. 

11) Discussion

  1. “These findings suggest that the role of CK2 in regulating MVM life cycle is independent of the localization of NS1 “. Please describe the role of CK2 in MVM life cycle since I am not sure all the readers are familiar with that.

Response: Thank you for the suggestion. We have provided a description of CK2’s role in parvovirus life cycle, its connection to the DDR pathways and how it might be connected in the revised Discussion section (Page 11).  

Round 2

Reviewer 3 Report

The authors significantly improved their manuscript. However, several issues still stand, which should be addressed in a further revision.

In particular

1) The authors have provided analysis of inhibitors activity (sup Fig.1). This is good, however they tested activity and not specificity. As a very minimum, they should check the specificity of iATM and iATR. This means that for both conditions phosphoATM and phosphoATR foci should be quantified (similarly to what has been done for caffeine)

2) The authors quantified NS1 delocalization in Fig, 2. However, as far as I understood they simply quantified NS1 fluorescence intensity. This means their quantification is influenced by protein expression levels. I would recommend a normalization of NS1 fluorescence over the irradiated area over total NS1 fluorescence. BY expressing data as a ratio rather than as an absolute number, quantification should be much more stable. Also, this analysis would be complementary and not alternative to the one already presented. Each cell should have an irradiated/non irradiated Fluorescence ratio, which will allow calculation of mean +/- SD relative to each condition.

3) the authors state that at late times post infection pCHK1 is not present in irradiated area, whereas NS1 delocalization is unaffected. Unfortunately, here (see panel 1C, bottom) it appears that only a minor fraction of NS1 as compared to early stages is delocalized. A quantitative analysis should be added. I suggest performing a ratio analysis as suggested above, to be shown alone the intensity analysis the authors perform in Figure 2. 

4) The x axes of the panels showing quantification of NS1 delocalization are labelled as "unit distance along...". It would be much more appropriate to express distances in microns. 

5) Quantification for the number of ADAR bodies (Fig3F) is missing. Also, it looks like in untreated cells NS1 largely localizes with gH2AX, whereas upon ATR inhibition gH2AX NS1 is mainly surrounded by them. This is evident from the pictures, as well as from the color RGB profiles shown on the right. The authors also refer to this phenotype "VM APAR bodies form in distinct pockets surrounded or directly next to gH2AX [26].", This should be mentioned in the results and thoroughly discussed. 

6) Most panel labels in new Fig.3 are messed up. In the text they refer to panel F (Confocal imaging) describing the results of Western blotting and so on so forth. These should be carefully checked throughout the manuscript. 

7) The authors keep on referring to protein kinase CK2 as to "casein kinase 2": Legend to Supplementary Fig.2 and Discussion. This should be checked throughout the manuscript and corrected

8) Something funny must have happened here in the discussion "Previous work investigating APAR body formation has shown that while at lower resolutions NS1 appears to be directly associated with gH2AX, at higher resolution, MVM APAR bodies form in distinct pockets surrounded or directly next to gH2AX [26]. However, at higher resolution, MVM APAR bodies form in distinct pockets surrounded or directly next to gH2AX [26]. ".

9) Figure 3H I think the label "iATR" is partially hidden by a white box. Maybe double check.

Again, I must stress this is a very nicely conducted work which, once revised as recommended, deserves publication. However, I need to ask the Authors to pay special attention to all the small details such as those highlighted in points 6-9. Since proofread a manuscript it is extremely time consuming, I could not systematically double check everything. I can therefore only hypothesize many other small inaccuracies could be hidden somewhere.

Author Response

1) The authors have provided analysis of inhibitors activity (sup Fig.1). This is good, however they tested activity and not specificity. As a very minimum, they should check the specificity of iATM and iATR. This means that for both conditions phosphoATM and phosphoATR foci should be quantified (similarly to what has been done for caffeine)

Response: Thank you for the suggestion. We have provided additional quantification of phosphor-ATM and phosphor-ATR foci counts during both ATM and ATR inhibition to show the specificity of the inhibitors (Supplemental Figure S1).

2) The authors quantified NS1 delocalization in Fig, 2. However, as far as I understood they simply quantified NS1 fluorescence intensity. This means their quantification is influenced by protein expression levels. I would recommend a normalization of NS1 fluorescence over the irradiated area over total NS1 fluorescence. BY expressing data as a ratio rather than as an absolute number, quantification should be much more stable. Also, this analysis would be complementary and not alternative to the one already presented. Each cell should have an irradiated/non irradiated Fluorescence ratio, which will allow calculation of mean +/- SD relative to each condition.

Response: Thank you for the suggestion. We have quantified NS1 intensity along the striped region normalized to the total NS1 expression for at least 10 nuclei for all inhibitors used in Fig 2. The Results of this quantification are shown in Supplemental Figure S3 with the mean and SEM as well as the relevant statistical tests. We have updated the Materials and Methods section accordingly to reflect this analysis.

3) the authors state that at late times post infection pCHK1 is not present in irradiated area, whereas NS1 delocalization is unaffected. Unfortunately, here (see panel 1C, bottom) it appears that only a minor fraction of NS1 as compared to early stages is delocalized. A quantitative analysis should be added. I suggest performing a ratio analysis as suggested above, to be shown alone the intensity analysis the authors perform in Figure 2. 

Response: Thank you for the suggestion. Since U2OS cells are a non-synchronous system for MVM infection but ideal for laser micro-irradiation, we had intended for these assays to be qualitative. However, we have calculated the ratio of NS1 levels along the laser stripe to that of total nuclear NS1. These numbers are as follows: panel 1 (NS1): 0.404; panel 2 (MVM early): 0.293 and panel 3 (MVM late): 0.197. These numbers also corroborate our broad conclusions but need to be rigorously followed up using laser micro-irradiation experiments in a synchronous model system of MVM infection. We have included these ratios to the figure legend in Fig. 2.

4) The x axes of the panels showing quantification of NS1 delocalization are labelled as "unit distance along...". It would be much more appropriate to express distances in microns. 

Response: Thank you for your suggestion, we have modified the quantification of NS1 intensity graphs to reflect the distance in microns where applicable (Fig. 2)

5) Quantification for the number of ADAR bodies (Fig3F) is missing. Also, it looks like in untreated cells NS1 largely localizes with gH2AX, whereas upon ATR inhibition gH2AX NS1 is mainly surrounded by them. This is evident from the pictures, as well as from the color RGB profiles shown on the right. The authors also refer to this phenotype "MVM APAR bodies form in distinct pockets surrounded or directly next to gH2AX [26].", This should be mentioned in the results and thoroughly discussed. 

Response: Thank you for pointing this out. Quantification of the number of APAR bodies is an inaccurate representation of the efficiency of virus replication. Instead, we have rigorously quantified virus replication by western blots for NS1 and genome copies of the plus strand. Therefore, in this instance, the APAR body imaging is a qualitative visualization of where viral replication centers form in the presence of iATR relative to cellular DNA damage sites. We have described these findings in the results section and discuss them further in the Discussion section.

6) Most panel labels in new Fig.3 are messed up. In the text they refer to panel F (Confocal imaging) describing the results of Western blotting and so on so forth. These should be carefully checked throughout the manuscript. 

Response: We have corrected the figure callouts in the manuscript. Thank you for pointing this out.

7) The authors keep on referring to protein kinase CK2 as to "casein kinase 2": Legend to Supplementary Fig.2 and Discussion. This should be checked throughout the manuscript and corrected

Response: Thank you for identifying the additional instances of the mislabel. We have modified all references of Casein Kinase 2 to Protein Kinase CK2.

8) Something funny must have happened here in the discussion "Previous work investigating APAR body formation has shown that while at lower resolutions NS1 appears to be directly associated with gH2AX, at higher resolution, MVM APAR bodies form in distinct pockets surrounded or directly next to gH2AX [26]. However, at higher resolution, MVM APAR bodies form in distinct pockets surrounded or directly next to gH2AX [26]. ".

Response: Thank you for catching this typo. We have taken out the second sentence.

9) Figure 3H I think the label "iATR" is partially hidden by a white box. Maybe double check.

Response: This is intended to be labelled as “:iATR”. However, to remove confusion, we have modified the labels in Fig. 3H.